# Assessment of Bulbar Function in Adult Patients with 5q-SMA Type 2 and 3 under Treatment with Nusinersen

**DOI:** 10.3390/brainsci11091244

**Published:** 2021-09-20

**Authors:** Svenja Brakemeier, Benjamin Stolte, Andreas Thimm, Kathrin Kizina, Andreas Totzeck, Juan Munoz-Rosales, Christoph Kleinschnitz, Tim Hagenacker

**Affiliations:** Department of Neurology and Center for Translational Neuro- and Behavioral Sciences (C-TNBS), University Hospital Essen, Hufelandstr. 55, 45147 Essen, Germany; svenja.brakemeier@uk-essen.de (S.B.); benjamin.stolte@uk-essen.de (B.S.); andreas.thimm@uk-essen.de (A.T.); kathrin.kizina@uk-essen.de (K.K.); andreas.totzeck@uk-essen.de (A.T.); Juan.munoz-rosales@uk-essen.de (J.M.-R.); christoph.kleinschnitz@uk-essen.de (C.K.)

**Keywords:** spinal muscular atrophy, ASO, ALSFRS-R, SSQ, swallowing

## Abstract

The antisense oligonucleotide nusinersen has been shown to improve trunk and limb motor function in patients with spinal muscular atrophy (SMA). Bulbar dysfunction, which is regularly present in SMA, is not captured by standard motor scores, and validated measurement instruments to assess it have not yet been established. Data on whether and how bulbar function changes under gene-based therapies in adult SMA patients are also unavailable. Here, we present data on the course of bulbar dysfunction assessed prospectively before nusinersen treatment initiation and 6 and 14 months later in 23 adult SMA patients using the Sydney Swallow Questionnaire (SSQ) and the bulbar subscore of the Amyotrophic Lateral Sclerosis Functional Rating Scale Revised (ALSFRS-R). While no improvement in bulbar scores was observed under treatment with nusinersen, the absence of a decline still implies a therapeutic effect of nusinersen on bulbar dysfunction. The results of this study aim to contribute to a standardized assessment of bulbar function in adult SMA patients, which may show therapeutic effects of gene-based therapies that are not evident from standard motor scores.

## 1. Introduction

The hereditary autosomal recessive neuromuscular disorder, 5q-associated spinal muscular atrophy (SMA), causes progressive weakness of the trunk, limb, ventilatory and bulbar muscles. The most common cause is a homozygous deletion or compound heterozygosity with deletion and point mutation in the survival motor neuron 1 (*SMN1*) gene [1], resulting in reduced expression of the corresponding SMN protein and consequent degeneration of spinal motor neurons [2]. The most important disease modifier is the *SMN2* gene, which differs in only five base pairs from *SMN1*, leading to the expression of a truncated SMN protein. The copy number of *SMN2* correlates inversely with disease severity [3]. Onset age and the best motor milestone achieved characterize different historical disease phenotypes, with types 0–4 corresponding to decreasing motor function [4]. Weakness and atrophy primarily affect axial and proximal muscle groups but also bulbar muscles due to the involvement of brainstem motor nuclei. Feeding disability is prevalent in approximately one-third of SMA patients [5]. In the “pretherapeutic” era, SMA type 1 patients suffered from symptoms during early infancy and most often died before the age of two without mechanical ventilation due to severe bulbar and respiratory impairment, type 2 patients typically achieved the ability to sit, and type 3 patients achieved the ability to walk independently and had a normal life expectancy [6]. Nonetheless, the natural course of the disease is chronic and progressive with a loss of motor milestones over time [7].

The antisense oligonucleotide nusinersen was the first causative treatment option for SMA. It alters splicing of the premRNA of *SMN2*, leading to increased formation of full-length SMN protein [8]. Nusinersen leads to relevant motor improvements, which has been shown in two large pivotal studies enrolling children with type 1 and type 2 SMA [9,10] and with increasing real-world evidence in adult SMA patients [11,12,13]. To objectively evaluate motor improvement, established motor function scales such as the Hammersmith Functional Motor Scale Expanded (HFMSE) and the Revised Upper Limb Module (RULM) have been used. However, these scores represent only trunk and extremity function, and bulbar function in adult SMA patients under therapy with nusinersen has not been comprehensively assessed to date. Impaired bulbar function, which manifests as difficulties in mouth opening, chewing and swallowing, is especially evident in early and severely affected SMA patients, mostly SMA type 1 and 2 patients, with an increasing incidence with a longer disease course [14]. Several studies investigating particular bulbar function components such as the mandibular range of motion, including the maximal mouth opening, bite force, masticatory function and swallowing using clinical, electrophysiological and video fluoroscopic examinations, questionnaires and MRI scans found that these functions were reduced in SMA patients, particularly in those with type 1 and 2 disease [14,15,16]. However, measures of bulbar function in SMA, especially in adult patients, have not yet been validated and standardized.

With an evolving therapeutic landscape, a standardized assessment of bulbar function in addition to already established motor scores is essential to improve therapeutic outcome measures. The purpose of this study was to assess bulbar function in adult SMA patients receiving nusinersen.

## 2. Materials and Methods

### 2.1. Sample

The study was conducted at the Department of Neurology, University Medicine Essen, Germany. Data were collected prospectively from July 2017 until May 2021. Patients provided written informed consent prior to their inclusion in the study. The study was approved by the local ethics committee of the University of Duisburg Essen, Germany (approval number 18-8071-BO). Of a total of 57 SMA patients who had received at least 5 injections of nusinersen by May 2021, 22 patients with SMA type 2 and 3, documented bulbar dysfunction before treatment initiation and without a percutaneous enteroscopic gastrostoma were included in the analyses. All patients had molecularly confirmed 5q-SMA and had been on therapy for at least 6 months. A total of 19 patients had already received 7 injections of nusinersen at the time of data analysis, i.e., they had already been on therapy for 14 months. Nusinersen was administered intrathecally at 12 mg initially for saturation on days 1, 14, 28 and 63 and, then, consecutively at four-month intervals in accordance with the label [17].

### 2.2. Assessment of Bulbar Function

Bulbar function was assessed with bulbar function items of the Amyotrophic Lateral Sclerosis Functional Rating Scale Revised (ALSFRS-R) and using the Sydney Swallow Questionnaire (SSQ). Data collection was conducted during the patients’ visits for nusinersen injection before the first injection (T0, baseline), after 6 months of therapy (T1) and after 14 months of therapy (T2). Both assessment tools are questionnaire-based, and the German versions were used. The ALSFRS-R is an established and validated measurement tool for monitoring disability progression in patients with amyotrophic lateral sclerosis [18]. Three of 12 items—speech, salivation and swallowing—reflect bulbar function in this tool and, thus, represent the bulbar subscore of the ALSFRS-R. Each item can be assigned a score of 0–4, resulting in a total possible score of 0–12 for the bulbar subscore, with lower scores indicating more severe bulbar impairment. The SSQ is a patient-reported outcome measure primarily assessing complaints of oropharyngeal dysphagia and has been shown to be valid, reliable and sensitive for assessing neurogenic, structural and age-related swallowing complaints in several patient populations in its German version [19]. It consists of 17 questions addressing subjective swallowing difficulties, e.g., bolus consistency, duration of food intake and signs of penetration, aspiration and regurgitation. The questions are designed to cover three subscales of global swallowing function, physical impairments and swallowing-related quality of life. Answers are given on a visual analog scale of 100 mm in length such that an arithmetic mean of 0–100 can be calculated from all 17 questions, with higher scores indicating more severe bulbar impairment.

Additionally, HFMSE and RULM scores, as established measures of extremity motor function in SMA, were obtained at T0, T1 and T2. The HFMSE consists of 33 items addressing activities of daily living, each rated on a 3-point scale (0–2). The maximum total score is 66 [20]. The RULM assesses upper extremity function and consists of 19 items, all but one of which is rated on a 3-point scale (0–2). The maximum total score is 37 [21]. Both scores are particularly used for type 2 and 3 SMA patients, with higher scores indicating better motor function, and have been validated even in young adults [22].

### 2.3. Statistical Analyses

For pre-post comparisons of the bulbar subscore of the ALSFRS-R and SSQ scores between T0 and T1 and between T0 and T2, a Wilcoxon signed rank test was performed. In secondary analyses, pre-post comparisons were also performed for the HFMSE and RULM. Descriptive statistics contain all available data at the respective time points, regardless of whether a complete dataset T0, T1 and T2 was available for a patient, other than in the pre-post comparisons. The results were visualized in boxplots with mean lines added. In subgroup analyses, separate pre-post comparisons were calculated for type 2 and type 3 SMA patients. Correlations for baseline values were calculated using Spearman’s rank correlation coefficient between the subscore of the ALSFRS-R and the SSQ score and between the two bulbar scores and the HFMSE and RULM scores. Alpha was set to ≤ 0.05. Bulbar function scores were correlated with SMN copy number and age in exploratory analyses.

## 3. Results

The demographic and clinical characteristics of the patients are given in Table 1.

No significant change in bulbar function measured by the bulbar subscore of the ALS-FRS-R was found between T0 and T1 (z = −0.302, *p* = 0.763, *n* = 15) or between T0 and T2 (z = −1.406, *p* = 0.160, *n* = 13). The mean bulbar subscores from the ALSFRS-R were 10.50 (standard deviation (sd) = 1.21, *n* = 16) at T0, 10.50 (sd = 1.32, *n* = 20) at T1 and 10.94 (sd = 0.97, *n* = 18) at T2. Five patients showed improved scores, three showed worsened scores, and seven showed unchanged scores from T0 to T1. From T0 to T2, the bulbar subscore on the ALSFRS-R was improved in five patients, worsened in two patients and remained unchanged in six patients. No significant change was found in the SSQ score between T0 and T1 (z = −0.210, *p* = 0.834, *n* = 13) or between T0 and T2 (z = −0.392, *p* = 0.695, *n* = 12). The mean SSQ scores were 23.59 (sd = 18.85, *n* = 18) at T0, 24.12 (sd = 19.32, *n* = 15) at T1 and 19.95 (sd = 17.86, *n* = 13) at T2. Regarding the SSQ, from T0 to T1, improvements were reflected by lower scores in eight patients and decreases in five patients. From T0 to T2, six patients’ SSQ scores improved, and six patients’ scores worsened. The data distribution and results of the paired comparisons are shown in Figure 1. Measures of central tendency and dispersion of the bulbar scores and the HFMSE and RULM scores at T0, T1 and T2 are also listed in Table 2.

Additional subgroup analyses, in which we calculated the pre-post comparisons of the two bulbar scores for type 2 and type 3 SMA patients separately, did not show significant changes in bulbar function either and, therefore, did not reveal differences in treatment response. Moreover, among the treated patients with no documented bulbar dysfunction at treatment initiation that we did not include in our sample, no new bulbar impairments developed over a 14-month period, as could be observed for 23 SMA patients.

A significant positive correlation was found between the baseline bulbar subscore of the ALSFRS-R and baseline limb motor function measured by the RULM (r = 0.434, *p* = 0.043, *n* = 16). No significant correlation was found between the baseline SSQ score and baseline bulbar subscore of the ALSFRS-R (r = −0.304, *p* = 0.336, *n* = 12). A strong positive correlation was found between baseline HFMSE and baseline RULM scores (r = 0.925; *p* < 0.001, *n* = 22). Correlational analyses did not reveal associations of the bulbar or extremity motor scores with age and SMN2 copy number.

In secondary analyses, we found a significant improvement in the HFMSE score (z = −2.236, *p* = 0.025, *n* = 22) from T0 to T1. The difference in the HFMSE score from T0 to T2 was not significant (z = −1.248, *p* = 0.212, *n* = 18). No significant pre-post differences were found for the RULM score between T0 and T1 (z = −1.932, *p* = 0.053, *n* = 22) or between T0 and T2 (z = −0.256, *p* = 0.798, *n* = 18).

## 4. Discussion

No relevant improvement in bulbar function was found in adult SMA type 2 and 3 patients under nusinersen treatment for up to 14 months. Subgroup analyses from SMA type 2 and 3 patients revealed no differences in treatment response, which goes in line with findings from another study indicating that the best predictor of dysphagia is the individual’s current level of motor function instead of SMA type [23]. Bulbar function has never been investigated in a larger-sample study in adult SMA patients under nusinersen. This study contributes to the development of standardized measures of bulbar function in adult SMA patients.

More severely affected SMA patients were more likely to have bulbar functional impairment, as reflected by the higher proportion of SMA type 2 patients and low baseline mean HFMSE and RULM scores in our cohort. Both scores—the mean baseline HFMSE score and the mean baseline RULM score—were lower in our sample than the corresponding baseline scores from another large-scale study investigating nusinersen therapy in adult SMA patients [11]. With higher disease severity at baseline, less improvement in extremity motor function under nusinersen is observed [11,13]. This is one reason why our results cannot be used to predict a poor response to therapy on the basis of pre-existing bulbar dysfunction alone. Consistent with this, only a partial improvement in limb motor function was found in our sample under nusinersen treatment, which was reflected by an improvement in the HFMSE score from T0 to T1 without any change in the RULM score. The absence of improvement in bulbar scores under nusinersen in this study may be due to the already advanced stage of degenerative disease involving brainstem neurons, where bulbar function is no longer amenable to therapy. However, the positive correlation between the baseline bulbar subscore of the ALSFRS-R and the RULM score may suggest a common mechanism determining the progression of both extremity function and bulbar function. SMA is naturally a chronic progressive disease throughout adolescence and adulthood, as demonstrated by a constant decline in muscle strength and motor function measured by the HFMSE [24]. An estimated average decline in the HFMSE score of 0.5 points per year was found for type 2 and 3 SMA patients [6,25]. Reliable data on the natural history of bulbar function in SMA, especially regarding adult patients, are not available to a similar extent, which is why a comparison between the course of bulbar dysfunction in our sample and the natural disease course in untreated patients can only be drawn based on the following assumptions. The chronic progressive decline in extremity function in adult patients and bulbar dysfunction progression in SMA type 1 patients, who regularly undergo a switch from oral to alternative feeding forms during the natural disease course, imply a natural progressive course of bulbar dysfunction in adult SMA patients [23], which strengthens the assumption of a therapeutic effect of nusinersen on bulbar function in adult SMA patients, as evidenced by the absence of a further decline in bulbar scores. Recently, summarized data from the few existing studies investigating dysphagia outcomes under disease-modifying therapies in SMA corresponded to a total of seven studies, which mostly only referred to SMA type 1 patients and were performed with small sample sizes and heterogeneous outcome measures [23]. Data on adult patients are scarce. In a pilot study by Kruse et al. (2020), two adult SMA type 2 patients showed improvements in maximum bite force measured using a piezoelectric force sensor, coinciding with the application of the initial loading doses of nusinersen. Long-term improvement was not observed over the course of one year of therapy [26]. In a recently published study on the treatment effects of nusinersen in adult SMA patients older than 17 years, patients reported qualitative improvement of bulbar symptoms, without this being objectified [27]. Additionally, data on swallowing ability under risdiplam in infants imply preserved swallowing function and oral feeding ability throughout the first 12 months of therapy, indicating a relevant therapeutic effect of this gene-based therapy [28]. In an online survey on satisfaction with early nusinersen therapy, adult and pediatric SMA patients rated the effect of nusinersen on bulbar symptoms such as eating and talking similarly as satisfactory by about 65% [29].

The SSQ, as a patient-reported outcome measure, allows an individualized and patient-centered assessment of swallowing dysfunction severity. The questionnaire is easy to understand and contains questions regarding subjective functional impairment from oropharyngeal dysphagia, whereas symptom-specific quality of life is not as thoroughly addressed [19]. Comparable validated dysphagia-specific questionnaires in German-speaking countries include the SWAL-QOL [30] and the MD Anderson Dysphagia Inventory [31], which primarily assess swallowing-related quality of life, and the Eating Assessment Tool-10 [32], which serves as a screening rather than a monitoring tool. Regarding the SSQ, a recent work on screening for oropharyngeal dysphagia in adult patients with neuromuscular diseases showed that the individual item scores for most questions included in the SSQ correlated with the total SSQ score in SMA [33]. Other reports on the application of the SSQ specifically in SMA are not yet available. Similarly, no data on the bulbar subscore of the ALSFRS-R in SMA are available, whereas this tool is widely used for assessment in amyotrophic lateral sclerosis. The SMA Functional Rating Scale (SMA-FRS), which is a modification of the ALS-FRS, was developed specifically for SMA but excludes bulbar function [34]. The calculation of subscores of the ALSFRS-R is well established and allows a more precise assessment [35]. The ALSFRS-R score is a patient-reported outcome measure and can be obtained by the investigator as well as by self-assessment in standard practice. While the SSQ includes numerous questions regarding swallowing only, the bulbar subscore of the ALSFRS-R assesses speech, salivation and swallowing. Thus, it measures bulbar function more broadly.

Previous studies suggest that in addition to swallowing alone, masticatory motor function and strength, mandibular range of motion [14,15,16] and head posture also account for bulbar impairments in SMA, implying that the SSQ may insufficiently cover the whole bulbar spectrum in SMA. Of note, in our sample, the baseline bulbar subscore of the ALSFRS-R correlated with the baseline RULM score, in contrast to the SSQ score. The fact that we found no association between the baseline bulbar scores can be viewed in the context of the abovementioned differences between the two scores.

Whether improvements in bulbar scores under nusinersen would show after a follow-up period longer than 14 months remains unclear. Long-term observations up to 3 years of children with SMA type 2 or 3 under nusinersen imply sustained improvements in motor function throughout this period [36]. Preliminary exploratory long-term data from 2–25-year-old type 2 and 3 SMA patients receiving risdiplam also revealed improvements in the HFMSE and RULM scores from months 12 to 24 [37]. Nonetheless, even stabilization of bulbar function and, thus, the absence of further decline in bulbar scores should be considered a therapeutic effect.

When evaluating our study results, certain limitations should be considered. First, a sample size of 22 patients and recruitment in a single center limit the representativeness and statistical power of the study. Additionally, ALSFRS-R and SSQ scores were not available for all patients at all time points, although they were for most patients. Next, given the limited observation period of a maximum of 14 months, our study does not provide data on possible longer-term treatment effects. Future studies with longer observation periods are necessary in this regard. However, both the sample size and the observation period were comparable to those in other studies in this field of rare disease research. The scores used to assess bulbar function have not been validated specifically for SMA. They were selected based on the abovementioned considerations and were found to be suitable measurement instruments for our investigation.

Our findings raise new research questions. It should be aimed to validate the data collected by means of questionnaires with more objective methods. The video fluoroscopic examination using flexible endoscopic evaluation of swallowing (FEES) is an established tool that could be used for this purpose but is also clearly more invasive. Potential non-invasive methods may include imaging, such as real-time MRI-assessment of swallowing, which has been used in patients with inclusion-body myositis [38], but whose evaluation seems very complex. Our idea, though, was to investigate and establish easy-to-use scores that allow for frequent follow-up assessments without much effort for patients and examiners. Ideally, a direct comparison of the course of bulbar function under gene therapy and in untreated patients could be drawn. However, in the context of growing and increasingly adopted therapeutic options, generating data for this is difficult.

## 5. Conclusions

Evaluating bulbar function in addition to extremity motor function in adult SMA patients under disease-modifying therapies may reveal additional meaningful treatment effects, especially in severely affected patients. While we observed preservation of bulbar function without a further significant decline under nusinersen, future studies with larger samples are needed to detect possible improvements in bulbar function under gene-based therapies. For assessing bulbar function in adult SMA patients, the bulbar subscores of the ALS-FRS-R and the SSQ may complement each other well. Further research on this topic may lead to more individualized therapeutic decision making in the treatment of SMA. 

## Figures and Tables

**Figure 1 brainsci-11-01244-f001:**
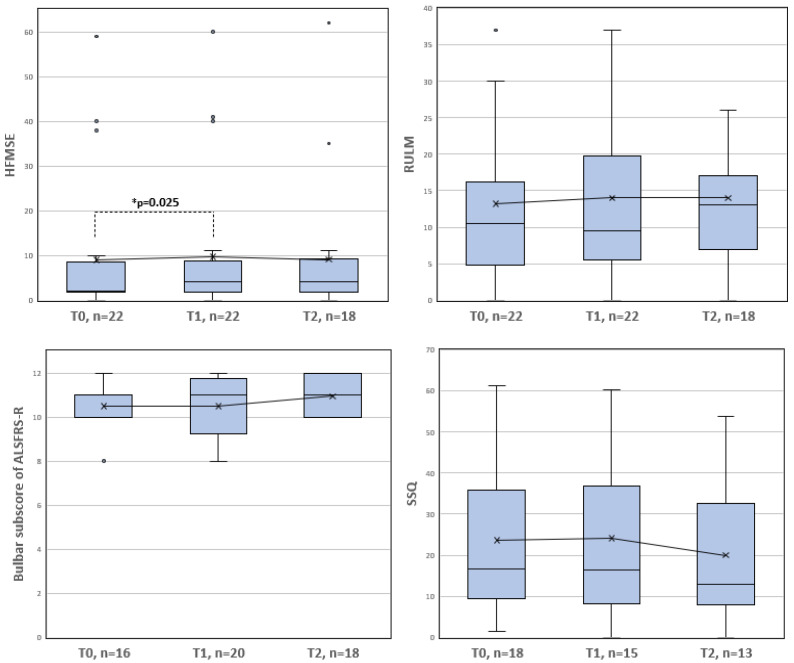
Measures of central tendency and dispersion and the mean bulbar and extremity motor scores at T0, T1 and T2. The boxes are defined by the upper and lower quartiles. The median is marked as a continuous line in the box. The length of the whiskers is limited to a maximum of 1.5 times the interquartile range. Data points outside this range are marked as circles. Crosses represent the respective arithmetic means. Significant differences based on a significance level of α = 0.05 and calculated with the Wilcoxon signed rank test are indicated by an asterisk *.

**Table 1 brainsci-11-01244-t001:** Demographic and clinical characteristics of all patients (*n* = 22).

Sex	Male (%)	Female (%)		
	13 (59)	9 (41)		
**Presence of Spondylodesis**	**Yes (%)**	**No (%)**		
	10 (45)	12 (54)		
**SMA type**	**2 (%)**	**3 (%)**		
	12 (54)	10 (45)		
**SMN 2 Copy Number**	**3 (%)**	**4 (%)**	**5 (%)**	**≥6 (%)**
	14 (64)	6 (27)	1 (4.5)	1 (4.5)
**Age**	**Mean (sd)**	**Range**	**Minimum**	**Maximum**
	38.5 (14.2)	52	20	72
**Symptom Duration in Years**	**Mean (sd)**	**Range**	**Minimum**	**Maximum**
	34.3 (11.9)	44	14	58

**Table 2 brainsci-11-01244-t002:** Descriptive statistics for the bulbar subscore of the ALSFRS-R, SSQ, HFMSE and RULM at T0 (baseline), T1 (6 months) and T2 (14 months).

	N	Mean (sd)	Minimum	Maximum	Lower Quartile	Median	Upper Quartile
**Bulbar subscore of ALSFRS-R (0–12)**							
**T0**	16	10.50 (1.21)	8.0	12.0	10.0	11.0	11.0
**T1**	20	10.50 (1.32)	8.0	12.0	9.25	11.0	11.75
**T2**	18	10.94 (0.97)	10.0	12.0	10.0	11.0	12.0
**SSQ (0–100)**							
**T0**	18	23.59 (18.85)	1.47	61.18	9.48	16.69	35.95
**T1**	15	24.12 (19.32)	0.0	60.29	8.23	16.05	36.88
**T2**	13	19.95 (17.86)	0.0	53.82	8.00	12.81	32.48
**HFMSE (0–66)**							
**T0**	22	8.95 (15.64)	0.0	59.0	1.75	2.0	8.5
**T1**	22	9.68 (15.92)	0.0	60.0	1.75	4.0	8.75
**T2**	18	9.06 (15.46)	0.0	62.0	1.75	4.0	9.25
**RULM (0–37)**							
**T0**	22	13.23 12.04)	0.0	37.0	4.75	10.5	16.25
**T1**	22	14.0 (12.11)	0.0	37.0	5.5	9.5	19.75
**T2**	18	14.0 (10.57)	0.0	37.0	7.0	13.0	17.0

## Data Availability

The data presented in this study are available on request from the corresponding author.

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
