# Peer review of "Assessment of Bulbar Function in Adult Patients with 5q-SMA Type 2 and 3 under Treatment with Nusinersen"

_brainsci, 2021, doi:10.3390/brainsci11091244_

Round 1

Reviewer 1 Report

The authors investigated bulbar dysfunction in patients with SMA using ALSFRS-R bulbar subscore and the SSQ questionnaire before and after treatment with nusinersen. They included patients with SMA I (one patient), SMA II and III. For this reason the title is misleading, it is correct they investigated adult subjects but including many patients with childhood disease onset. It would be more reasonable to exclude the single patient with SMN type I and to write in the tithe “Assessment of bulbar function in SMA type II and II patients with….”

Introduction, the second paragraph merges clinical features (described in the first paragraph) and treatment rational, better to remove the repeated clinical information.

Methods, important to specify that SSQ was validated to German. I recommend to remove the single patient with SMA type I, and after describing the results for the whole group to analyze if the subgroups including patients with type II and type III show a different response to treatment. It is incorrect to compare 17 patients (only) with ALSFRS-R scoring at T0 with 21 at T1 and 19 at T2, the same group of 17 should be compared in the different time sets without including more patients.

Statistical analysis, correct median lines

One limitation of this study (to be added in discussion) is that not all patients underwent ALSFRS-R and SSQ scoring, moreover we observe a significant dropout for the SSQ questionnaire.

Tables are not very good and table 3 is unnecessary, the few significant findings could be added in the text. The p values should be added in table 2. The figure 1 is of poor quality, in particular concerning HFMSE box. The p value for the significant change is missing, p<0.05 is not enough.

Discussion (and conclusion) can be shortened. It would be important to focus more on SMA III, due to the poor available information in the literature. The absent correlation between ALSFRS-R bulbar subscore and the SSQ questionnaire results from the former to analyze other bulbar features like dysarthria. This part of the discussion can very short.

Reviewer 2 Report

The paper describes bulbar dysfunction in SMA patients assessed before and after 6/14 months of nusinersen treatment. The topic is timely and of clinical interest, yet the study needs some clarifications, additional analyses and more in-depth discussion to make it suitable for publication.

From 57 treated patients, 23 with bulbar dysfunction were selected for this study. It would be of interest to document how many of the other 34, despite nusinersen treatment, later developed bulbar symptoms. I suggest to add this data to the results section. Perhaps therapy can prevent swallowing difficulties to later arise? And would the authors dare go so far as to claim that pre-existing bulbar dysfunction could predict limited response to nusinersen? In this respect, the discussion should go past pure narration of the (limited) clinical effects reported and dare to draw conclusions and propose more new research questions.

There are some questions regarding methodology that I would like the authors to answer. Both the Sydney Swallow Questionnaire and the ALSFRS‐R are self-reported measures, commonly used and valid. Yet, aren’t there objective measurable clinical parameters that could have been used alongside? And why was Wilcoxon signed rank test chosen when more than 2 groups needed to be analyzed? The study claims to ‘aim to contribute to a standardized assessment of bulbar function’, but such a strategy needs to be more clearly developed in the text.

Can a more detailed comparison be provided with measures expected during the natural course of disease? The study showed no significant improvement of bulbar function, and notes that ‘absence of decline’ is potential proof of benefit. Please include the expected timeline of bulbar dysfunction in the reported cohort in the discussion. Also, only modest effects on motor function were reported, with HFMSE scores improved significantly from T0 to T1 only. Is this comparable to other reported patients with similar baseline symptoms? The positive correlations that were observed between bulbar and motor function, were these present at all time points? And could certain patients be categorized as non-responders? Were any differences noted between SMA type 2 and 3?

Refer and analyze also Chen et al., 2020 and Duong et al., 2021 for comparisons of reported bulbar dysfunction improvement after nusinersen treatment. Also, perhaps data has been released from the Nurture study?

To evaluate possible conflict of interest, please provide all forms of support received by the authors from Biogen and partners.

Round 2

Reviewer 1 Report

This reviewer is pleased with this version

Reviewer 2 Report

The manuscript was adequately revised.